# An Efficient and Robust Framework for Approximate Nearest Neighbor Search with Attribute Constraint

**Mengzhao Wang**
Hangzhou Dianzi University
`wmzssy@yeah.net`

**Lingwei Lv**
Hangzhou Dianzi University
`llw@hdu.edu.cn`

**Xiaoliang Xu**[*]
Hangzhou Dianzi University
`xxl@hdu.edu.cn`

**Yuxiang Wang**
Hangzhou Dianzi University
`lsswyx@hdu.edu.cn`

**Qiang Yue**
Hangzhou Dianzi University
`yq@hdu.edu.cn`

**Jiongkang Ni**
Hangzhou Dianzi University
`hananoyuuki@hdu.edu.cn`

## Abstract

This paper introduces an efficient and robust framework for hybrid query (HQ) processing, which combines approximate nearest neighbor search (ANNS) with attribute constraint. HQ aims to find objects that are similar to a feature vector and match some structured attributes. Existing methods handle ANNS and attribute filtering separately, leading to inefficiency and inaccuracy. Our framework, called native hybrid query (NHQ), builds a composite index based on proximity graph (PG) and applies joint pruning for HQ. We can easily adapt existing PGs to this framework for efficient HQ processing. We also propose two new navigable PGs (NPGs) with optimized edge selection and routing, which improve the overall ANNS performance. We implement five HQ methods based on the proposed NPGs and existing PGs in NHQ, and show that they outperform the state-of-the-art methods on 10 real-world datasets (up to $315\times$ faster with the same accuracy).

## 1 Introduction

Approximate nearest neighbor search (ANNS) is a crucial problem in data science and AI applications [27, 43, 9, 47]. For example, in a paper-retrieval system based on ANNS, a user wants to find papers that are most similar to her query text. As Fig. 1(a) shows, the system converts each paper's unstructured text and the query text into feature vectors in a high-dimensional space. Then it uses a vector index to perform ANNS and obtain papers with similar content. Many effective ANNS methods have been developed to balance query efficiency and accuracy [18, 19, 36, 40, 11].

However, ANNS does not support many real-world scenarios where users want to find objects with similar feature vectors to the query object and meets the given attribute constraint (e.g., topic, venue, and publish_year of a paper) [51, 41]. We call this *a hybrid query (HQ)* [51, 54, 60, 32, 50, 41, 42]. For instance, in Fig. 1(b), a user wants to find some recent papers from top-tier conferences related to her research interest. She can form a HQ by providing a descriptive text of her interest and two paper attributes (e.g., NeurIPS and 2022). However, the traditional ANNS shown in Fig. 1(a) only finds papers with similar content, and it cannot guarantee matching attribute constraint. Therefore, many have tried to add attribute filtering (AF) on top of ANNS to answer a HQ [54, 51, 60].

Vearch [29] uses ANNS to find candidates that match the feature vector, and then applies AF to get the final results [32]. This strategy can work with other ANNS libraries, such as SPTAG [37], NGT [59], and Faiss [17]. On the other hand, Alibaba AnalyticDB-V (ADBV) [54] employs product

---

[*]Corresponding author

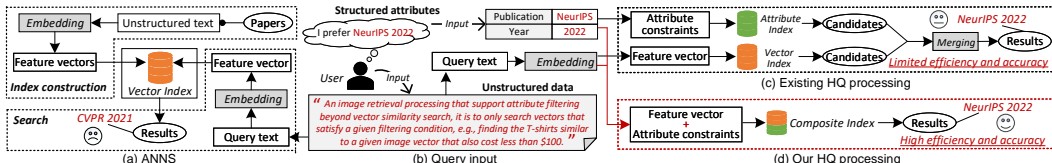

Figure 1: An example of different paper-retrieval schemes, including (a) ANNS and (c–d) HQ. ANNS retrieves results with semantically similar content, but it does not check the attribute constraint. A HQ processing obtains results that match both the feature vector and attribute constraints.

quantization (PQ)[28] to create multiple query plans. One of them is to do AF first and then ANNS to return the final results. This method is scalable and suitable for most use cases [51]. Milvus [38] partitions the dataset into subsets based on common attributes. This way, it can quickly select the subsets that meet the query's attribute constraints, and only perform ANNS on them instead of the whole dataset. This improves the search efficiency [51].

These solutions have a common idea: they do AF and ANNS separately in different orders (see Appendix A for details). No matter what order they use, the whole query pipeline can be simplified as Fig. 1(c), where two subquery systems (for AF and ANNS) are built independently. They can answer a HQ, but they are not designed for it, which affects their efficiency and effectiveness.

## 1.1 Limitations of Existing Methods

**L1: Two indexes need to be kept at the same time.** Fig. 1(c) shows that existing methods need both the attribute index and vector index for two separate subquery systems. This takes more memory space and adds more logic (e.g., *merging*) to make sure two indexes are consistent and query results are correct [54]. Updating different indexes at the same time is hard when the data changes often.

**L2: Extra computational cost from two separate pruning strategies.** For existing methods, pruning is done based on attribute index and vector index separately, which increases computational cost. For example, suppose that the vector index in Fig. 1(c) is a proximity graph (PG, we define it in **Def. 4**) [52], where each vertex is an object's feature vector. Then, some vertices near the query object's feature vector but not matching the attributes can make the search follow wrong paths and return wrong answers to a HQ. We prune these wrong answers in the attribute-filtering phase, but the computational cost for returning them in the ANNS phase is unnecessary and inefficient.

**L3: Query results rely on the merging of the candidates from both subquery systems.** As Fig. 1(c) shows, we need to merge the top-$k$ candidates $C_1$ and $C_2$ from both subquery systems to get the final results $R = C_1 \cap C_2$. But usually we have $|R| \ll k$. This is because existing methods do ANNS and AF separately, so each system can only return candidates that meet one type of constraint. To make $|R|$ bigger, we need to make $C_1$ and $C_2$ bigger, which makes both systems slower [60].

**L4: Existing methods are not friendly for PGs.** Recent methods, like ADBV [54] and Milvus [51], use a "first AF, then ANNS" strategy to answer a HQ; they only use PQ [28] for ANNS. But many studies have shown PG is $10\times$ faster than PQ for ANNS [5, 33], so we need to find a way to use PG in HQ. A simple way is to replace PQ-based vector index with PG-based one, but it has some problems: (1) In the "first AF, then ANNS" strategy, we want to find similar vectors from a dynamic space of the vectors filtered by attributes. It is not efficient to build a vector index for these filtered vectors in runtime. (2) Even worse, if we list all possible combinations of attribute values and prebuild a vector index for each one offline (like [56]), it would take too much memory space, e.g., $m^n$ different indexes for $n$ attributes where each attribute has $m$ values.

To sum up, most of these limitations happen because existing methods answer HQ in a "decomposition-assembly" model, where a HQ is split into two subqueries that process separately and then combine the results. This motivates us to present a new general framework for HQ that works well with existing PGs. It has the well-designed *composite index* and *joint pruning* modules to support ANNS with attribute constraint, instead of keeping two separate indexes and doing pruning separately.

## 1.2 Our Solution and Contributions

To the best of our knowledge, we are the first to present a **N**ative **H**ybrid **Q**uery framework (NHQ) for ANNS with attribute constraint in a fused way, not the "decomposition-assembly" model used by

existing methods. As Fig. 1(d) shows, our framework first combines feature vectors and attributes in a well-designed composite index (for **L1**); then it prunes the search space on the composite index by considering both the feature vector and attribute constraints (for **L2**); and finally, it gets the final top-$k$ results directly without a merging operation (for **L3**). Our NHQ is a general framework, where the composite index is based on PG, so existing PGs can easily work with NHQ (for **L4**) and it also supports custom-optimized PGs. Our main contributions are:

- We introduce NHQ, a general framework for ANNS with attribute constraint (**§3**). NHQ works well with existing PGs, and can make them handle HQ only by a lightweight modification.
- We design two navigable PGs (NPGs) with optimized edge selection and routing strategies (**§4**), which have better efficiency-vs-accuracy trade-off than basal PGs.
- We use the proposed NPGs and existing PGs in our NHQ framework, and get several new HQ methods that beat the state-of-the-art competitors on all datasets (up to 315× faster, **§5.3**).
- We show our methods' better effectiveness, efficiency, and scalability on 10 real-world datasets, compared with state-of-the-art HQ frameworks and methods (**§5**).

## 2 Preliminaries

**Definition 1** *Object Set. An object set is defined as a set $\mathcal{S} = \{e_0, \ldots, e_{n-1}\}$ of size $n$. For each object $e \in \mathcal{S}$, (1) its features are represented as a high-dimensional vector, denoted by $\nu(e)$, (2) $e$ has a set of attributes denoted by $\{a_0, \ldots, a_{m-1}\}$ and $e.a_i$ indicates the value of attribute $a_i$ of $e$. Moreover, we define the feature vector set of all objects in $\mathcal{S}$ as $\mathcal{X} = \{\nu(e)|e \in \mathcal{S}\}$.*

**Example 1** *An object set could refer to different types of data, e.g., images, papers. When we specify each object $e$ as a paper, it carries two types of information: one is the implicit semantics behind the text, which is usually represented as a feature vector $\nu(e)$ by neural network, e.g., BERT [13]; and another is the explicit attributes, e.g., $\{\text{venue}, \text{topic}\}$, such as $e.\text{venue} = \text{"NeurIPS"}$.*

Given an object set $\mathcal{S}$ and a user-input query object $q$, many approaches have been studied to retrieve the most similar objects to $q$ from $\mathcal{S}$ by considering feature vectors' distance alone [15, 36, 16]. In the following, we first introduce ANNS, then formally define the HQ that we study.

For an object $e \in \mathcal{S}$ and its feature vector $\nu(e) \in \mathcal{X}$, we write $\nu(e) = [\nu(e)^0, \nu(e)^1, \ldots, \nu(e)^{d-1}]$, where $\nu(e)^i$ is the value of $\nu(e)$ on the $i$-th dimension. We focus on the high-dimensional case where $d$ is from hundreds to thousands. For any two objects $e, o \in \mathcal{S}$ with feature vectors $\nu(e), \nu(o) \in \mathcal{X}$, we can measure their similarity with different methods, such as Euclidean distance [19] and Cosine similarity [36]. Euclidean distance is the most popular method [33], which is in Eq. 1.

$$\delta(\nu(e), \nu(o)) = \sqrt{\sum_{i=0}^{d-1}(\nu(e)^i - \nu(o)^i)^2} \tag{1}$$

**Definition 2** *NNS [18, 20, 45]. Given an object set $\mathcal{S}$ and a query object $q$ with the feature vector $\nu(q)$, the NNS aims at obtaining the top-k objects from $\mathcal{S}$ whose feature vectors are closest to $\nu(q)$.*

In Def. 2, the exact top-$k$ objects (denoted by $T$) hold that

$$T = \arg\min_{T \subseteq \mathcal{S}, |T|=k} \sum_{e \in T} \delta(\nu(e), \nu(q)) \quad . \tag{2}$$

Exact NNS on a large $\mathcal{S}$ is not feasible because it takes too much computation [52]. So, an ANNS is more realistic, as it balances accuracy and efficiency with a vector index [52]. Let $T$ be the exact top-$k$ objects from Def. 2 and $R$ be the approximate top-$k$ objects from an ANNS method. We can use recall rate $Recall@k$ to measure the search accuracy of the approximate method:

$$Recall@k = \frac{|R \cap T|}{k} \quad . \tag{3}$$

A bigger $Recall@k$ means more accurate results from ANNS.

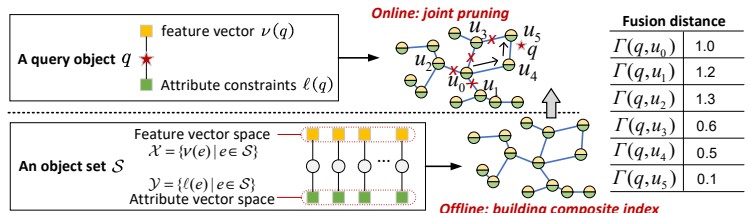

Figure 2: NHQ framework overview.

To return the query results fast, ANNS makes a vector index based on the feature vector. We can classify ANNS methods into four types by how they make the index: quantization [28, 23]; tree [46, 2]; hashing [22, 24]; and proximity graph (PG)[36, 19]. Many works [33, 19] have shown that the PG-based methods have better efficiency vs accuracy trade-off.

**Definition 3** *Hybrid Queries (HQ). Given an object set $\mathcal{S}$ and a query object $q$ with a feature vector $\nu(q)$ and a set of attributes $\{a_0, \ldots, a_{m-1}\}$ of size $m$, the HQ returns the top-$k$ objects from $\mathcal{S}$, denoted by $T$. The objects in $T$ satisfy two conditions: ① attribute filtering (AF): they have the same attributes as $q$—that is, for any $e \in T$, $\forall i = 0, 1, \cdots, m-1$, $e.a_i = q.a_i$; and ② ANNS: their feature vectors are the closest to $\nu(q)$ among those that meet ①.*

It is also too slow to answer an exact HQ on a large $\mathcal{S}$. So, current research works on an approximate HQ processing [51, 54] and the query accuracy is measured by Eq. 3 for a returned $R$ from the approximate method. HQ is an extended ANNS with attribute constraint [51]. But it is a challenging problem, because all existing methods have accuracy and efficiency issues (**L1-L4 in §1.1**).

## 3   NHQ Framework

We begin by defining proximity graph (PG) and then introduce the NHQ framework based on PG.

**Definition 4** *PG. Given an object set $\mathcal{S}$, we define the PG of $\mathcal{S}$ w.r.t. an distance threshold $B$ as a graph $G = (V, E)$ with the vertex set $V$ and edge set $E$. ① For each vertex $u \in V$, it corresponds to an object $e \in \mathcal{S}$. ② For any two vertices $u_i$ and $u_j$ from $V$, if $u_i u_j \in E$, we have $\delta(\nu(u_i), \nu(u_j)) \leq B$, where $\nu(u_i)$ and $\nu(u_j)$ is the feature vectors of objects $u_i$ and $u_j$, respectively.*

Let $N(u_i)$ be the neighbor set of $u_i$. Our idea is to build a PG that uses both feature vectors and attributes as a composite index. In the index, the neighbors of $u$ may have the same attributes as $u$. We can use Eq. 1 to measure the feature vector distance, but we need to quantify the attributes and define a distance metric function [55]. Then, we can fuse the feature vector distance and the attribute distance into a fusion distance to compare the objects. Therefore, we can directly prune the vertices that have either dissimilar feature vectors or different attributes with the fusion distance.

**Fusion distance.** Given an object set $\mathcal{S}$ with feature vectors $\mathcal{X}$ (see Fig. 2, bottom left), we can use different encoding methods [3, 44, 48] to encode the attributes of each object $e \in \mathcal{S}$. Ordinal encoding works well for structured attributes [53]. We use ordinal encoding $\ell(.)$ to get an attribute vector $\ell(e) = [\ell(e)^0, \cdots, \ell(e)^{m-1}]$ for each object $e$, where $\ell(e)^i$ is $e.a_i$'s encoded value (e.g., {NeurIPS, NLP} can be encoded as $[1, 3]$ by $\ell(.)$). Then, we get attribute vectors $\mathcal{Y} = \{\ell(e) | e \in \mathcal{S}\}$.

We can measure the distance between two feature vectors $\nu(e_i)$ and $\nu(e_j)$ in $\mathcal{X}$ by using Eq. 1. We can measure the distance between two attribute vectors $\ell(e_i)$ and $\ell(e_j)$ in $\mathcal{Y}$ by using this formula:

$$\chi(\ell(e_i), \ell(e_j)) = \sum_{k=0}^{m-1} \phi(\ell(e_i)^k, \ell(e_j)^k) \quad, \tag{4}$$

where $\phi(\ell(e_i)^k, \ell(e_j)^k)$ is

$$\phi(\ell(e_i)^k, \ell(e_j)^k) = \begin{cases} 0 & \ell(e_i)^k = \ell(e_j)^k \\ 1 & \ell(e_i)^k \neq \ell(e_j)^k \end{cases} \quad. \tag{5}$$

In Eq. 4–5, $m$ is the number of dimensions in $\ell(e_i)$, and $\ell(e_i)^k$ is the value on the $k$-th dimension. The smaller this distance is, the more similar the attribute vectors are.

We can fuse the feature vector distance $\delta(\nu(e_i), \nu(e_j))$ and the attribute vector distance $\chi(\ell(e_i), \ell(e_j))$ into a single distance $\Gamma(e_i, e_j)$ for objects $e_i$ and $e_j$ by using this formula:

$$\Gamma(e_i, e_j) = w_\nu \cdot \delta(\nu(e_i), \nu(e_j)) + w_\ell \cdot \chi(\ell(e_i), \ell(e_j)) \quad , \tag{6}$$

where $w_\nu$ and $w_\ell$ are distance weights. The smaller this distance is, the more similar the objects are in both feature vectors and attributes. Eq. 6 is a simple and practical way to combine two different distances. It also makes it easy to build a composite index on top of existing PGs, because we just need to change the distance measure from Eq. 1 to Eq. 6. For example, if we set $w_\nu = 1$ and $w_\ell = 0$, we get $\Gamma(e_i, e_j) = \delta(\nu(e_i), \nu(e_j))$, which is the same as building a PG based on feature vector. If we set $w_\nu = 0$ and $w_\ell = 1$, we get $\Gamma(e_i, e_j) = \chi(\ell(e_i), \ell(e_j))$, which is a PG based on attribute. So, we can find the best balance between two distances by adjusting $w_\nu$ and $w_\ell$.

**Weight configuration.** We found that $w_\nu = 1$ and $w_\ell = \delta(\nu(e_i), \nu(e_j))/m$ give the best query performance for most datasets and algorithms (see Appendix U). These weights do not depend on the dataset, but only on the feature vector distance $\delta(\nu(e_i), \nu(e_j))$ of two objects and the attribute vector dimension $m$. The idea is to fine-tune the feature vector distance with the attribute distance. For example, if $e_i$ and $e_j$ have the same attributes, that is, $\chi(\ell(e_i), \ell(e_j)) = 0$, we keep the feature vector distance as it is (i.e., $\Gamma(e_i, e_j) = \delta(\nu(e_i), \nu(e_j))$). If $e_i$ and $e_j$ have completely different attributes, that is, $\chi(\ell(e_i), \ell(e_j)) = m$, we double the feature vector distance to get $\Gamma(e_i, e_j) = 2 \cdot \delta(\nu(e_i), \nu(e_j))$. In general, we have $\delta(\nu(e_i), \nu(e_j)) \leq \Gamma(e_i, e_j) \leq 2 \cdot \delta(\nu(e_i), \nu(e_j))$.

**Composite index.** We build a composite index based on Algorithm 1 (see Fig. 2, bottom right). We start with $V = \mathcal{S}$, so each object $e \in \mathcal{S}$ is a vertex $u \in V$ in $G$, and $E = \emptyset$. Then, we add an edge $u_i u_j \in E$ between two vertices $u_i, u_j \in V$ if $\Gamma(u_i, u_j) \leq B'$, where $B'$ is a fusion distance threshold. Note that our composite index does not incur extra space cost, as shown in Lemma 1.

**Lemma 1** *The composite index and the ordinary PG have the same index size for an object set $\mathcal{S}$.*

---

**Algorithm 1:** Building Composite Index $(\mathcal{S})$

**Input:** Object set $\mathcal{S}$
**Output:** *Composite Index $G = (V, E)$*
$V \leftarrow \mathcal{S}, E \leftarrow \emptyset$
**forall** $u_i \in V$ **do**
    **forall** $u_j \in V \setminus \{u_i\}$ **do**
        **if** $\Gamma(u_i, u_j) \leq B'$ **then**
            $E = E \cup \{u_i u_j\}$

return $G = (V, E)$

---

**Theorem 1** *Let $\Omega_{min}$ and $\Omega_{max}$ be the minimum and maximum distances between feature vectors on $\mathcal{S}$. Suppose we have at least $|N(e)|$ objects with the same attributes as $e$ for any object $e \in \mathcal{S}$. Then, ① for any vertex $u$ and its neighbor $o$ in the composite index, we have $\Omega_{min} \leq \Gamma(u, o) \leq \Omega_{max}$; ② for any vertex $u$ and its neighbor $o$ in the ordinary PG, we have $\Omega_{min} \leq \Gamma(u, o) \leq 2 \cdot \Omega_{max}$.*

Theorem 1 shows that our composite index has a smaller fusion distance bound.

**Joint pruning.** Given a query object $q$, we use a composite index $G$ and a seed set $P$ (usually chosen randomly from $V$ [52]) to find the approximate top-$k$ objects based on Algorithm 2. We follow these steps (see Fig. 2, top right): ① *Initialization.* We use a visited vertex set $C$ to store the vertices for search expansion and a result set $R$ of size $k$ to store the current query results. We set both sets to $P$ at first. ② *Search expansion.* We take out the vertex $u_i$ with the smallest $\Gamma(q, u_i)$ from $C$ as the next visited vertex, and then add $N(u_i)$ to $C$. ③ *Query results update.* We update $R$ with the better vertices in $N(u_i)$. For any vertex $u_j \in N(u_i)$ and a vertex $u_r \in R$ that is the farthest from $q$, we replace $u_r$ with $u_j$ in $R$ if $\Gamma(q, u_j) < \Gamma(q, u_r)$. This means $u_j$ is more similar to $q$ in both feature vectors and attributes than $u_r$. We repeat ② and ③ until $R$ does not change, then we return $R$ as the approximate top-$k$ objects.

---

**Algorithm 2:** Joint Pruning $(G, q, P)$

**Input:** *Composite index $G = (V, E)$,*
      *query object $q$, seed set $P$*
**Output:** Result set $R$
candidate set $C \leftarrow P$; result set $R \leftarrow P$
**while** $R$ *is updated* **do**
    $u_i \leftarrow \arg\min_{u_i \in C} \Gamma(q, u_i)$;
    $C = C \setminus \{u_i\}$
    $N(u_i) \leftarrow$ the neighbors of $u_i$;
    $C = C \cup N(u_i)$
    **forall** $u_j \in N(u_i)$ **do**
        $u_r \leftarrow \arg\max_{u_r \in R} \Gamma(q, u_r)$
        **if** $\Gamma(q, u_j) < \Gamma(q, u_r)$ **then**
            $R = R \setminus \{u_r\}$;
            $R = R \cup \{u_j\}$

return $R$

---

**Example 2** *We show an example of joint pruning for finding the nearest object (i.e., top-1) to the query object q in Fig. 2 (top right). We start with a random vertex as the seed, i.e., $P = \{u_0\}$,*

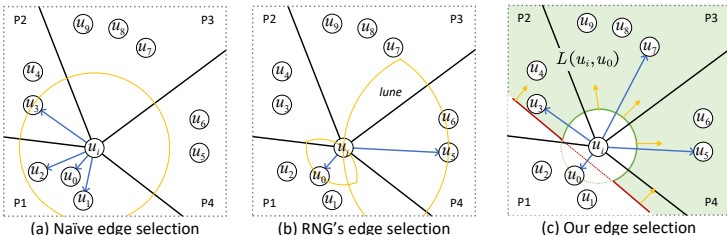

Figure 3: Different edge selection strategies for $u_i$ (we assume $u_i$ has at most four neighbors).

*and set $C = \{u_0\}$ and $R = \{u_0\}$. Then, we expand the search space with $u_0$'s neighbors and get $C = \{u_1, u_2, u_3, u_4\}$. We update $R$ with $u_4$ because $\Gamma(q, u_4) = 0.5$ is smaller than $\Gamma(q, u_0) = 1.0$. We continue to expand the search space with $u_4$'s and $u_5$'s neighbors and get $R = \{u_5\}$. Since no vertex $u_j \in N(u_5)$ has a smaller distance than $\Gamma(q, u_5)$, we return $u_5$ as the approximate top-1 nearest object. We prune the vertices $u_1$, $u_2$, and $u_3$ from the search space.*

**Theorem 2** *Given a query object $q$, the composite index guarantees at least the same $Recall@k$ of HQ as the ordinary PG. (see Appendix H for the proof)*

Theorem 2 states that our composite index has a higher or equal $Recall@k$ than the ordinary PG.

## 4 Navigable PG Algorithm

To form the composite index, we can deploy a specific PG in NHQ and change its original distance measure to the fusion distance (Eq. 6). This gives the PG the ability to handle HQ effectively. However, current PGs have limitations that affect NHQ's performance with the PG-based composite index. Therefore, we present two new navigable PGs (NPGs) in this section. We optimize the *edge selection* and *routing* strategies for building and searching on a PG, respectively.

### 4.1 Edge Selection

Edge selection is a crucial step for building a PG. It determines the neighbors of each object $e$ in an object set $\mathcal{S}$. Different strategies build different structures for a PG, which affect its search performance [19]. Existing PGs use two factors for edge selection: *distance between two vertices* (D1) and *distribution of all vertices* (D2) [52]. Early PGs like NSW [35] and KGraph [14] only use D1 and connect each vertex with some of its nearest neighbors (Fig. 3(a)). However, this can cause redundant computations and reduce search efficiency [34]. Recent PGs also use D2 with RNG's edge selection, which diversifies the neighbors' direction [33]. However, RNG's edge selection still fails to diversify the neighbors well (Fig. 3(b)). Due to the space limitation, we explain this in Appendix I.

**Our edge selection.** We design a new edge selection strategy that uses D1 and D2. It connects $u_i$ with one nearest neighbor in each area of $u_i$ (P1–P4 in Fig. 3(c)). We define the *landing zone* of $u_i$ and one of its neighbors $u_j \in N(u_i)$, which is the area where only the vertices in it can be added to $N(u_i)$. We then describe our edge selection strategy based on the landing zone.

**Definition 5** *Landing zone. The landing zone $L(u_i, u_j)$ of a vertex $u_i$ and one of its neighbors $u_j \in N(u_i)$ is an area defined by $H(u_i, u_j) \setminus B(u_i, \delta(\nu(u_i), \nu(u_j)))$. Here, $H(u_i, u_j)$ is the half space with $u_i$ that is split by the perpendicular bisector $U(u_i, u_j)$ of the line between $u_i$ and $u_j$. And $B(u_i, \delta(\nu(u_i), \nu(u_j)))$ is the hypersphere centered at $u_i$ with radius $\delta(\nu(u_i), \nu(u_j))$.*

**Example 3** *As Fig. 3(c) shows, in a two dimensional space, $U(u_i, u_0)$ is a perpendicular bisector of the line connecting $u_i$ and $u_0$ (i.e., the red line), $H(u_i, u_0)$ is located on the upper side of $U(u_i, u_0)$, and $B(u_i, \delta(\nu(u_i), \nu(u_0)))$ is the area enclosed by the green circle. Therefore, the landing zone $L(u_i, u_0)$ formed by $u_i$ and $u_0$ is the green shaded region (i.e., $H(u_i, u_0) \setminus B(u_i, \delta(\nu(u_i), \nu(u_0)))$).*

To find the areas without any neighbor of $u_i$, we intersect the landing zones of all its existing neighbors (i.e., $\bigcap_{u_j \in N(u_i)} L(u_i, u_j)$). Then, we add the closest vertex (w.r.t. $u_i$) within this area into $N(u_i)$, making the neighbors more diverse. We get $N(u_i)$ for each vertex $u_i \in V$ as follows (Algorithm 3).

① *Candidates acquisition*. We obtain a set of $l$ candidate neighbors for $u_i$ from $(V \setminus \{u_i\})$, denoted by $C(u_i)$. We can use random sampling or an extra index to get $C(u_i)$ (e.g., we get $C(u_i)$ based

on NSW [35] and KGraph [14] in §4.3). We ensure that $l \geq R$, where $R$ is the maximum number neighbors for $u_i$ (i.e., $|N(u_i)| \leq R$). ② *Neighbors initialization.* We sort the elements in $C(u_i)$ in ascending order of distance to $u_i$. We initialize $N(u_i)$ with $u_i$'s nearest candidate neighbor $u_t$ from $C(u_i)$ and remove $u_t$ from $C(u_i)$. ③ *Neighbors update.* We select the nearest vertex $u_p$ of $u_i$ from $C(u_i)$, and add it to $N(u_i)$ if it is in the intersection area (i.e., $\bigcap_{u_j \in N(u_i)} L(u_i, u_j)$) of the landing zones of $u_i$ and its existing neighbors. We repeat this process until $C(u_i) = \emptyset$ or $|N(u_i)| = R$.

Our strategy ensures that $u_i$'s neighbors are diverse in terms of the areas where $C(u_i)$ is located. This is because we add $u_p$ to $N(u_i)$ from a different area each time. We have the following theorem.

**Theorem 3** *The angle between any two neighbors is not less than $\pi/3$. (See Appendix J for the proof)*

**Example 4** *In Fig. 3(c), assuming $N(u_i) = \{u_0\}$ and $C(u_i) = \{u_3, u_4, \cdots, u_9\}$, we include $u_3$ in $N(u_i)$ because it lies within $L(u_i, u_0)$ (the green region). Next, we add $u_5$ to $N(u_i)$ since it belongs to $L(u_i, u_0) \cap L(u_i, u_3)$. We continue this process until each new area has one neighbor of $u_i$. Consequently, in the search procedure, we can efficiently route to the query object $q$ by utilizing the neighbors of $u_i$ present in the same area as $q$.*

---

**Algorithm 3:** Edge Selection $(u_i, l, R)$

**Input:** Vertex $u_i$, constants $l$ and $R$
**Output:** Neighbor set $N(u_i)$
candidate set $C(u_i) \leftarrow l$ candidate neighbors
$u_t \leftarrow \arg \min_{u_t \in C(u_i)} \delta(\nu(u_i), \nu(u_t))$
$N(u_i) \leftarrow N(u_i) \cup \{u_t\}$;
$\quad C(u_i) \leftarrow C(u_i) \setminus \{u_t\}$
**while** $C(u_i) \neq \emptyset$ or $N(u_i) < R$ **do**
    $u_p \leftarrow \arg \min_{u_p \in C(u_i)} \delta(\nu(u_i), \nu(u_p))$
    $C(u_i) \leftarrow C(u_i) \setminus \{u_p\}$
    **if** $u_p \in \bigcap_{u_j \in N(u_i)} L(u_i, u_j)$ **then**
        $N(u_i) \leftarrow N(u_i) \cup \{u_p\}$

return $N(u_i)$

---

**Complexity.** To obtain $C(u_i)$, the time complexity depends on the methods used, so we skip this term in our analysis. The time complexity of sorting $C(u_i)$ is $O(l \cdot \log(l))$. For each $u_p \in C(u_i)$, we check at most $|N(u_i)|$ times ($|N(u_i)| \leq R$) to see if $u_p$ is in $L(u_i, u_j)$ for each $u_j \in N(u_i)$. So, our edge selection gets the final $N(u_i)$ with no more than $l \cdot R$ checks. Thus, the time complexity of our edge selection on the vertex set $V$ is $O(l \cdot (R + \log(l)) \cdot |V|)$.

## 4.2 Routing

Routing is a crucial process for searching on a PG. It determines a path from the start vertex to the result vertex that matches the query. The routing strategy affects both the efficiency and accuracy of the search [39]. A recent study [58] splits routing into two stages: *the far stage* (S1) and *the close stage* (S2), based on how far they are from the query object.

**Our routing.** Algorithm 4 illustrates our random two-stage routing strategy. In S1, we randomly select $\lceil R/h \rceil$ neighbors from $N(u_i)$ (where $1 \leq h \leq R$, and $R$ is the maximum out-degree), and calculate their distances to the query object $q$. This enables us to quickly approach the neighborhood of $q$. In S2, we evaluate the distances of all neighbors in $N(u_i)$ to $q$, following a similar approach to the greedy search [19]. The transition from S1 to S2 occurs when S1 reaches a local optimum, indicated by no further updates to the result set $R$. In S2, we continue updating $R$ by checking all neighbors of the visited vertex. The process terminates when $R$ no longer receives any updates.

**Theorem 4** *In the worst case, our routing has the same $Recall@k$ as the current greedy search [19].*

---

**Algorithm 4:** Routing $(G, q, P)$

**Input:** PG $G = (V, E)$, query object $q$, seed set $P$
**Output:** Result set $R$
candidate set $C \leftarrow P$; result set $R \leftarrow P$
**forall** $\{S1, S2\}$ **do**
    **while** $R$ *is updated* **do**
        $u_i \leftarrow \arg \min_{u_i \in C} \delta(\nu(q), \nu(u_i))$;
        $C = C \setminus \{u_i\}$
        S1: $M \leftarrow \lceil R/h \rceil$ random neighbors of $u_i$
        S2: $M \leftarrow$ all neighbors of $u_i$
        $C = C \cup M$
        **forall** $u_j \in N(u_i)$ **do**
            $u_r \leftarrow \arg \max_{u_r \in R} \delta(\nu(q), \nu(u_r))$
            **if** $\delta(\nu(q), \nu(u_j)) < \delta(\nu(q), \nu(u_r))$
            **then**
               $R = R \setminus \{u_r\}$; $R = R \cup \{u_j\}$
    $C \leftarrow C \cup R$;
return $R$

---

According to Theorem 4 (See Appendix L for the proof), our random strategy in S1 does not reduce the accuracy because most vertices do not require distance calculation from the query in S1 and we can recover a small number of possibly missed vertices in S2.

**Complexity.** Previous works [36, 19, 58] show that the time complexity of greedy search on a state-of-the-art PG (e.g., HNSW [36], NSG [19]) is $O(R \cdot \log(|V|))$, where $R \ll |V|$ is the maximum number of neighbors per vertex and $\log(|V|)$ is roughly the average routing path length. In our routing, we use $l_1$ and $l_2$ to denote the average routing path lengths in S1 and S2, respectively; then we have $l_1 + l_2 = \log(|V|)$. Thus, our routing's time complexity is $O((\lceil R/h \rceil) \cdot l_1 + R \cdot l_2)$.

### 4.3 NPG with Our Edge Selection and Routing

We present two NPGs, $NPG_{nsw}$ and $NPG_{kgraph}$, that use our edge selection and routing strategies on two basal PGs: NSW [35] and KGraph [14]. See Appendix M for implementation details.

## 5 Experiments

### 5.1 Experimental Setting

Table 1: Statistics of real-world datasets.

| Dataset | Dimension | # Base | # Query | LID [33, 18] | Type |
|---|---|---|---|---|---|
| UQ-V | 256 | 1,000,000 | 10,000 | 7.2 | Video + Attributes |
| Msong | 420 | 992,272 | 200 | 9.5 | Audio + Attributes |
| Audio | 192 | 53,387 | 200 | 5.6 | Audio + Attributes |
| SIFT1M | 128 | 1,000,000 | 10,000 | 9.3 | Image + Attributes |
| GIST1M | 960 | 1,000,000 | 1,000 | 18.9 | Image + Attributes |
| Crawl | 300 | 1,989,995 | 10,000 | 15.7 | Text + Attributes |
| GloVe | 100 | 1,183,514 | 10,000 | 20.0 | Text + Attributes |
| Enron | 1,369 | 94,987 | 200 | 11.7 | Text + Attributes |
| Paper | 200 | 2,029,997 | 10,000 | - | Text + Attributes |
| BIGANN100M | 128 | 100,000,000 | 10,000 | 9.3 | Image + Attributes |

**Datasets.** We use ten real-world datasets, including one newly released dataset called *Paper*. They span various modalities, such as video, image, audio, and text. We summarize their main characteristics in Tab. 1. With the exception of the *Paper* dataset, the remaining datasets solely consist of high-dimensional feature vectors without any original attributes. Therefore, we generate attributes for each object in these datasets using the similar method described in [51, 56]. For instance, in SIFT1M, we augment each image with attributes such as date, location, and size, thereby creating an object set that comprises both feature vectors and attributes.

**Compared methods.** We compare our HQ methods with seven existing ones that have been used in many high-tech companies. **ADBV** [54] is a cost-based HQ method proposed by Alibaba. It optimizes IVFPQ [28] for ANNS. **Milvus** [38, 51] divides the object set through frequently used attributes, and deploys ADBV [54] on each subset. **Vearch** [29, 32] is developed by Jingdong, which implements the HQ working off Strategy B. **NGT** [59] is a ANNS library released by Yahoo Japan, which answers a HQ to conduct attribute filtering atop the candidates recalled by NGT (Strategy B). **Faiss** [17] is a ANNS library developed by Facebook, which answers a HQ based on IVFPQ [28] and Strategy A. **SPTAG** [37] is a PG-based ANNS library from Microsoft, which answers HQ on Strategy B. **Filtered-DiskANN** [21] proposes two optimizations based on DiskANN: FilteredVamana and StitchedVamana. FilteredVamana connects vertices with shared attributes. StitchedVamana builds separate graph indexes for each filter and overlays them. **NHQ-$NPG_{nsw}$** and **NHQ-$NPG_{kgraph}$** are our HQ methods based on NHQ framework integrating two NPGs.

**Metrics.** We measure the search efficiency by *queries per second* (*QPS*), which is the number of queries ($\#q$) divided by the search time ($t$), i.e., $\#q/t$. We use the *Recall* rate to evaluate the search accuracy, which is defined by Eq. 3. Unlike ANNS, hybrid query also requires attribute constraints in Eq. 3, i.e., the elements in $R \cap T$ must have the same attributes as the query object.

**Implementation setup.** All codes are written in C++, and are compiled by g++ 6.5. All experiments are conducted on a Linux server with an Intel(R) Xeon(R) Gold 6248R CPU at 3.00GHz, and a 755G memory. We use 64 threads to build all the indexes in parallel. We use a single thread for search, which is a common setting in related work [19, 18]. We report the average results from three trials.

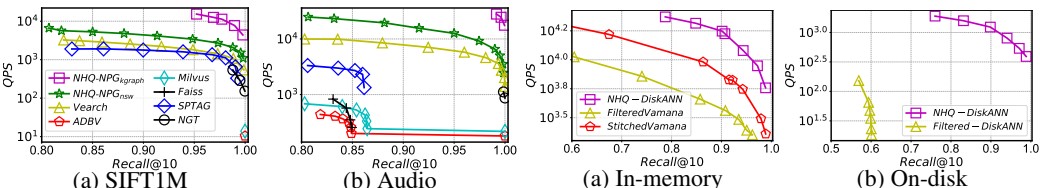

Figure 4: HQ performance.

Figure 5: Comparison with new baselines.

## 5.2 HQ Performance

Fig. 4 shows that our methods outperform existing methods in terms of *QPS* vs *Recall* trade-off on all datasets. For example, NHQ-$NPG_{kgraph}$ achieves two to three orders of magnitude higher *QPS* than others when *Recall@10* > 0.99. To execute Filtered-DiskANN, we only test single-attribute queries in Fig. 5. To eliminate other factors, we implement NHQ on DiskANN (NHQ can be easily extended to the current graph index), named NHQ-DiskANN. We keep the same parameters in DiskANN. From the results, NHQ-DiskANN outperforms Filtered-DiskANN, in both memory and disk versions.

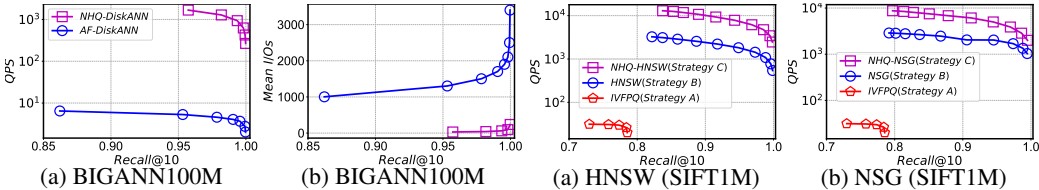

Figure 6: HQ performance on larger dataset.

Figure 7: Comparison of different strategies.

## 5.3 Scalability on Larger Dataset

To test the scalability of NHQ, we implement HQ on the state-of-the-art disk-resident PG, DiskANN [27], using "first ANNS, then AF" strategy (AF-DiskANN) and our NHQ (NHQ-DiskANN). Both implementations have the same hyper-parameters for DiskANN. Fig. 6(a) and (b) show the *QPS* and *Mean I/Os* results for different *Recall* on BIGANN100M dataset. NHQ-DiskANN consistently beats AF-DiskANN by a large margin. For *QPS*, NHQ-DiskANN is 315× faster than AF-DiskANN when Recall@10=0.95, because NHQ-DiskANN reduces expensive random disk I/Os. For example, to get a recall of 0.95, AF-DiskANN needs 1,303 disk I/Os, while NHQ-DiskANN only needs 30 I/Os, saving about 97.7% of disk I/Os. This shows the scalability of NHQ to handle larger dataset.

## 5.4 Ablation Study

**Validation of NHQ framework.** We test the universality of our NHQ (Strategy C) by applying the HQ model with "decomposition-assembly" to different PGs (including HNSW [36] and NSG [19]) based on "first ANNS, then AF" (Strategy B in Appendix A). We also use IVFPQ [28] in "first AF, then ANNS" (Strategy A in Appendix A) following state-of-the-art implementation [54, 51] because Strategy A does not support PG (**L4** in **§1.1**). We integrate HNSW and NSG into NHQ to form NHQ-HNSW and NHQ-NSG, respectively. As Fig. 7 shows, NHQ outperforms other strategies on different PGs, and keeps stable *QPS* advantage on different datasets. Due to IVFPQ's limitations, the HQ based on Strategy A have low accuracy, e.g., $Recall@10 < 0.8$ on SIFT1M.

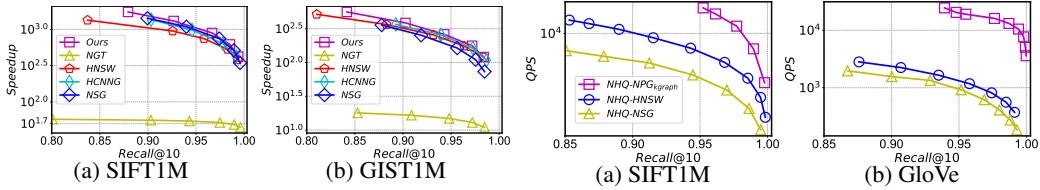

Figure 8: Effect of our edge selection strategy.

Figure 9: Effect of different PGs.

**Effect of our edge selection strategy.** We compare our edge selection strategy with four existing ones: NGT [59], HNSW [36], HCNNG [39], and NSG [19] on SIFT1M and GIST1M datasets using a recent evaluation framework [52]. All competitors use the same routing inference and distance function. We measure the *Speedup-Recall* metric, where *Speedup* is relative to brute force. Fig. 8

shows that our strategy outperforms the others. For example, at Recall@10=0.9 on SIFT1M, our strategy achieves 1.1×, 1.4×, and 25.9× speedup over HCNNG/NSG, HNSW, and NGT, respectively.

Table 2: Comparison of different routing strategies.

|        | Speedup (vs. HNSW) | Index size (MB) | Memory cost (MB) |
|--------|--------------------|-----------------|------------------|
| HNSW   | 1.00               | **790**         | **3,871**        |
| HCNNG  | 1.02               | 803             | 3,884            |
| TOGG   | 1.20               | 807             | 3,888            |
| FINGER | **1.30**           | 2,150           | 11,625           |
| Ours   | 1.21               | **790**         | **3,871**        |

**Effect of our routing strategy.** We compare our routing strategy with three existing ones: TOGG [58], FINGER [8], and HCNNG [39] within the HNSW index framework with consistent parameters. Tab. 2 shows the speedup of each optimized strategy over the original HNSW at Recall@10 = 0.9 on SIFT1M. All optimized strategies are faster than the original HNSW, indicating the benefit of optimizing the routing procedure. Our strategy has the lowest storage cost, as it does not require extra structures. Moreover, it delivers a significant improvement in search performance.

**Effect of different PGs under NHQ.** We compare the performance of three PGs on NHQ: HNSW [36], NSG [19], and our $NPG_{kgraph}$. We call them NHQ-HNSW, NHQ-NSG, and NHQ-$NPG_{kgraph}$ respectively. Fig. 9 shows their results on SIFT1M and GloVe. NHQ-$NPG_{kgraph}$ outperforms the others, especially on the harder GloVe dataset.

# 6 Discussion

**Memory overhead.** NHQ has a higher memory cost than Faiss when considering their best trade-offs. This is a common drawback of graph-based methods compared to PQ-based methods in the ANNS community. This is mainly because graph-based methods build an extra proximity graph index (stored as an adjacency list) to speed up the online search process. While graph-based methods have a higher storage cost, they achieve a significantly better trade-off between accuracy and efficiency and have become the mainstream algorithms in most vector databases (such as Milvus [51]). Our NHQ framework enhances the ability of current graph-based methods to handle ANNS + AF.

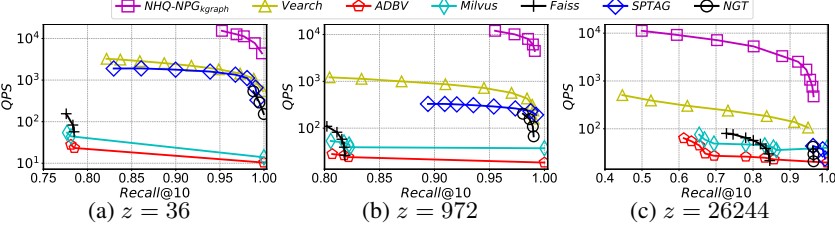

Figure 10: Hybrid query performance of different number of attribute combinations ($z$).

**Number of attribute combinations.** We conduct *QPS-Recall* comparisons across different attribute dimensions $\{3, 6, 9\}$ in Fig. 10. The corresponding number of attribute combinations (represented by $z$) is $\{36, 972, 26244\}$. We find that increasing $z$ can pose challenges for both the NHQ and existing frameworks. However, NHQ demonstrates higher speedup over existing frameworks in scenarios with a large $z$, indicating its potential advantage in handling diverse attribute sets efficiently.

# 7 Conclusion

In this paper, we address HQ, which is ANNS with attribute constraint. We present a NHQ framework that adapts to most existing PGs (including disk-resident PGs, e.g., DiskANN [27]) and optimizes them for HQ. We introduce two NPGs with enhanced edge selection and routing strategies that achieve better ANNS performance than original PGs. We integrate several PGs into NHQ to implement HQ methods that are up to 315× faster than the state-of-the-art alternatives. A possible future and ongoing work is to apply our framework to multimodal search to handle more complex query requirements.

# Acknowledgments

This work was supported by the Primary R&D Plan of Zhejiang (2021C03156 and 2023C03198) and the National NSF of China (62072149).

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
