# OpenReview forum: "An Efficient and Robust Framework for Approximate Nearest Neighbor Search with Attribute Constraint"
_NeurIPS.cc/2023/Conference — NeurIPS 2023 poster_

### Official Review · Reviewer_7xuq · 2023-07-05

**Soundness:** 4 excellent
**Presentation:** 4 excellent
**Contribution:** 3 good
**Rating:** 8
**Confidence:** 3

**Summary:**

The paper introduces a novel approximate nearest neighbor search framework which baked in attribute constraints via a single composite index compared to many existing two stage solutions. The framework is mainly relying on a newly proposed distance function that fuses both feature vector distances and attribute vector distances. Based on the new distance function, several proximity graph based ANNS methods have been developed. The paper presents an ablation study showing the merits of the methods. The experimental results shown on multiple popular landmark datasets outperform baseline methods.


**Strengths:**

The proposed fusion distance is an excellent idea that enables the opportunities of the single composite index construction as well as the advantages over the legacy two-stage models.

Each step of the method has been well-explained. In addition, most of the components have provided theoretical proof or support analysis.

ANN problems are often complicated in comparisons considering the nature of trade-offs among different factors including memory, accuracy, speed, etc. The paper has conducted sufficient experiments, ablation studies, and explanations of results. These strengthen the hypnosis and make the work overall to be solid.


**Weaknesses:**

Details about config or setups for some baselines are missing. For example, the parameters or usages of FAISS are not discussed. It makes it harder to understand why FAISS gets saturated at recall 80% as shown in Section 5.2. Is it possible that the number of probes in IVF has not increased sufficiently?

**Questions:**

There will be other challenges as the number of attributes increases. It may be worth discussing solutions if there are very high diversities of attributes among the data to be indexed. For example, different categories in product search may not share the same attributes (names). The total number of attributes could be considerably large, and it will also increase the attribute vector distance computation cost. The two stage approaches, e.g. AF + ANNS, are less likely to be constrained.

**Limitations:**

No limitations of the work are discussed in the paper.

Please refer to my questions and comments in Weakness and Questions.

---

> ### Author Rebuttal · Authors · 2023-08-09
>
> **W1: Experimental Details and Faiss Recall Issues**
>
> Thank you for your suggestions. Due to space limitations, we included the environment configuration and setups for the baselines in the appendix. We apologize for the omission of important details in the main paper. We'll improve the organization of the experiments section and include essential information such as experimental setup, datasets, and baseline descriptions in the main paper. Additionally, we'll also provide detailed elaborations in the appendix.
>
> In FAISS, we utilize the popular IVFPQ algorithm. In the indexing phase, IVFPQ first divides the database vectors into coarse clusters using k-means and then applies product quantization (PQ) to compress the vectors into compact codes. PQ divides each vector into $M$ subvectors and assigns each subvector to one of $K$ centroids in a codebook. The codebook is learned by performing *k*-means on the subvectors. The PQ code of a vector is the concatenation of the cluster IDs of its subvectors. This representation allows each vector to occupy $M$ bytes, where $M$ is typically 8 or 16 or 32. In the searching phase, IVFPQ first identifies the nearest coarse clusters to the query vector. It then scans the inverted lists of these clusters to find the nearest PQ codes to the query vector using asymmetric distance computation (ADC). ADC calculates the distance between the query vector and a PQ code by summing up the distances between each subvector of the query and the corresponding centroid of the PQ code. This approach avoids the need to decompress the PQ codes and reduces the memory access cost.
>
> Some key parameters of IVFPQ include:
>
> - The number of coarse clusters, which determines the number of inverted lists created and the number of clusters searched in each query. A larger number of clusters results in finer-grained partitioning but also increases computation and memory overhead.
> - The number of subvectors $M$ and the number of centroids $K$ in PQ, which determine the level of compression achieved and the amount of quantization error introduced. A larger $M$ or $K$ leads to higher accuracy but also increases storage and computation costs.
> - The number of nearest clusters to search for each query, which determines the number of inverted lists scanned and the number of PQ codes compared. A larger number yields higher recall but also increases computation cost.
>
> We determine the optimal parameters of IVFPQ for different datasets using the automatic parameter adjustment tool provided with FAISS. Due to compression errors, the recall rate of IVFPQ saturates at a certain value (e.g., ~80%), as widely observed in the literature [7] (refer to Fig. 2(a) of [7]), [8] (refer to Fig. 1 of [8]) and empirically verified in our experiments (Fig. 6(a)-(c) of the main paper). We would like to highlight that increasing the number of probes in IVF does not eliminate this limitation. Even when scanning all coarse clusters, the recall rate remains low. This limitation is a consequence of approximate distance computation.
>
> Thank you again for bringing up these points, and I hope this explanation clarifies the behavior of FAISS and its limitations in achieving higher recall rates.
>
> **Q1: High Diversities of Attributes**
>
> Thank you for your comments. It is indeed true that challenges arise as the number of attributes increases. To test the impact of varying diversities of attributes, we conduct QPS-Recall comparisons across different attribute dimensions {3, 6, 9}, as shown in Fig. 3 of the attached PDF file. Notably, the corresponding number of attribute combinations (represented by $z$) is {36, 972, 26244}. We find that increasing the number of attributes can pose challenges for both the NHQ and two-stage (e.g., ANNS+AF) frameworks. Here are some observations from our experiments:
>
> - Decreased QPS with increased number of attributes. The QPS decreases for all methods, including both NHQ and two-stage frameworks, as the number of attributes increases. For instance, when the number of attributes is 36, 972, and 26244, the QPS of the Vearch method is 1852, 725, and 106, respectively, for the same Recall@10 of 0.95. This decline in QPS highlights the fact that handling a larger number of attributes necessitates additional computational resources.
> - Higher speedup of NHQ over two-stage framework. NHQ demonstrates a higher speedup over the two-stage framework when dealing with a large number of attributes, despite the increased attribute vector distance computation cost in NHQ. For example, when the number of attributes is 36, 972, and 26244, with a Recall@10 of 0.95, NHQ achieves speedups of 9.0x, 17.2x, and 16.5x over Vearch, respectively. This indicates that NHQ is more efficient in scenarios with a large number of attributes compared to the two-stage framework. It's important to note that although NHQ may experience increased attribute vector distance computation costs, the two-stage framework is also affected by higher feature vector distance computation costs required to generate more intermediate results for attribute filtering. However, the feature vector distance computation cost is typically significantly higher than the attribute vector distance computation cost. As a result, the two-stage methods are more likely to be constrained by the number of attributes compared to NHQ.
>
> In summary, increasing the number of attributes introduces challenges for both NHQ and two-stage frameworks, leading to decreased QPS. However, NHQ demonstrates higher speedup over two-stage framework in scenarios with a large number of attributes, indicating its potential advantage in handling diverse attribute sets efficiently.

---

> > ### Comment · Reviewer_7xuq · 2023-08-19
> >
> > Thanks for the authors detailed answers. The ~80% recall plateaus of IVFPQ is most likely a consequence of the number of subspaces and codebook size configuration. Although it is not a major concern to me, I don't think it is a fair comparison and cannot agree that its already the optimal parameters for IVFPQ. In particular the memory cost of the proposed method is much higher than IVFPQ.

---

> > > ### Author Response · Authors · 2023-08-20
> > > **Response to Reviewer 7xuq**
> > >
> > > Thank you for your comments. We address each of your concerns in the following section.
> > >
> > > **IVFPQ Recall Plateau Issue**
> > >
> > > We agree that the ~80% recall plateau of IVFPQ is due to the configuration of the number of subspaces and codebook size. The trade-off in IVFPQ involves search accuracy, efficiency, and space cost. A larger number of subspaces and codebook size can improve search accuracy but decrease search efficiency. For instance, when the number of subspaces and the codebook size are the largest (referred to as IVFFlat in Faiss), we can achieve a recall rate of 100% by probing all clusters in IVF. However, this configuration may be slower than brute force search (refer to Fig. 6 in VLDB'19). Such low efficiency is impractical, so most research mainly focuses on the trade-off between search efficiency and accuracy in IVFPQ (VLDB'19, TPAMI'21, etc.). To achieve high efficiency, we set an appropriate number of subspaces and codebook size (not necessarily the largest possible) during the offline phase. We note that both the number of subspaces and codebook size are offline parameters. In the online search phase, we can only adjust the number of nearest clusters to probe for each query to achieve higher recall. Therefore, the recall rate of IVFPQ saturates at a certain value (which may vary across datasets) due to compression errors in the offline phase.
> > >
> > > We would like to clarify that we determined the optimal parameters for IVFPQ on different datasets using the automatic parameter adjustment tool provided by Faiss. Our evaluation indicates that the output parameters closely align with the optimal parameters obtained through grid search in most cases. Similar parameter settings have also been used in other literature (NeurIPS’20, NeurIPS’22, etc.) to evaluate IVFPQ.
> > >
> > > **Fair Comparison Issue**
> > >
> > > We acknowledge that the parameters for IVFPQ in our evaluation may not be optimal for achieving the highest recall or speedup. However, our paper focuses on the trade-off between accuracy and efficiency, not just accuracy or efficiency alone. Therefore, we configure the parameters to achieve the best trade-off for all methods, following the setting of related works (VLDB’19, NeurIPS’22, SIGMOD’23, etc.).
> > >
> > > We agree that NHQ, our proposed method, has a higher memory cost than IVFPQ when considering their best trade-offs. This is a common drawback of graph-based methods compared to PQ-based methods in the ANNS community (VLDB’19, NeurIPS’19). This is mainly because graph-based methods (such as HNSW) build an extra proximity graph index (stored as an adjacency list) to speed up the online search process (please see our analysis in the response to reviewer MN6o). However, graph-based methods are an order of magnitude more efficient than PQ-based methods in terms of queries per second (QPS) for a given recall, and this efficiency gap increases with data size (WWW’23). Therefore, while graph-based methods have a higher storage cost, they achieve a significantly better trade-off between accuracy and efficiency and have become the mainstream algorithms in most vector databases (such as Milvus, Pincone, AnalyticDB-V). Our NHQ framework enhances the ability of current graph-based methods to handle ANNS + AF.
> > >
> > > Thank you again for your constructive comments. We will add a discussion on the limitation of storage cost for NHQ. We believe there is still room for improvement in balancing storage and search performance (including accuracy and efficiency), and we plan to explore this in our future work.
> > >
> > >  **Reference:**
> > >
> > > VLDB'19: Fu, et al. Fast approximate nearest neighbor search with the navigating spreading-out graph.
> > >
> > > TPAMI'21: Fu, et al. High dimensional similarity search with satellite system graph: efficiency, scalability, and unindexed query compatibility.
> > >
> > > NeurIPS’20: Ren, et al. HM-ANN : Efficient billion-point nearest neighbor search on heterogeneous memory.
> > >
> > > NeurIPS'22: Chern, et al. Tpu-knn: K nearest neighbor search at peak flop/s.
> > >
> > > SIGMOD'23: Gao, et al. High-dimensional approximate nearest neighbor search: with reliable and efficient distance comparison operations.
> > >
> > > NeurIPS'19: Jayaram Subramanya, et al. Diskann: Fast accurate billion-point nearest neighbor search on a single node.
> > >
> > > WWW'23: Gollapudi, et al. Filtered-DiskANN: Graph algorithms for approximate nearest neighbor search with filters.

---

### Official Review · Reviewer_ffbB · 2023-07-05

**Soundness:** 3 good
**Presentation:** 2 fair
**Contribution:** 3 good
**Rating:** 7
**Confidence:** 5

**Summary:**

In this paper, authors tackle the problem of retrieving nearest neighbor items under constraints on attributes of the retrieved items. Each item is described by a feature vector and a set of discrete attributes.

Authors propose to use a distance function that uses weighted combination of feature vector based distance and discrete attribute based distance.

To support efficient retrieval for a given query and attribute constraints, authors use graph-based nearest neighbor search indices. The proposed hybrid distance function is used to build the graph in an offline step and to navigate the graph during test-time search.

Authors also propose two heuristics to improve graph construction and test-time graph navigation, and overall the proposed approach yields improvement over baselines.

Update: I am leaning towards accepting the paper and have updated my rating from  `5: borderline accept` to `7: accept` after reading clarifications from the authors.

**Strengths:**

- The proposed idea of using a hybrid distance function (which combines feature vector and attribute based distances) provides significant improvement over two-stage inference pipelines that either retrieve based on feature vectors and then filter based on attributes or first filter based on attribute and then search over filtered items using feature vector.
- The proposed approach outperforms popular baselines on a variety of datasets.

**Weaknesses:**

- Some missing baselines/ablations
    - The three main contribution of the paper are
        - a) New distance function for hybrid queries.
        - b) New algorithm for constructing graph over items.
        - b) New heuristic for efficiently navigating the graph at test-time.
    - The experiments clearly show the advantage of using the hybrid distance function over two-stage search (which separately performs nearest neighbor retrieval based on feature vector and then filters based on attributes).
    - But individual contribution of the proposed graph construction strategy and the proposed graph navigation strategy is not clear.
    - The proposed graph construction algorithm should be compared with NGT, HNSW, Munoz et al. (2019) while keeping every other design variable the same i.e. with the same test-time inference as well as with the same distance function.
    - Similarly, the proposed graph search method should be compared with existing methods for speeding up search such as TOGG (mentioned in the paper), Chen et al., (2023), Munoz et al., (2019).
- The presentation of the paper can be further improved.
    - Most algorithms are described in text but it would help to present them in Algorithm boxes.
    - While it is okay to use appendix for extra results, theorem proofs etc, I think some important details such as experiment setup, datasets and baseline description has also been moved to appendix. Reading the paper involved too many jumps between the main paper and the appendix. Authors could make the hybrid distance section more concise to make some space or move some extra results to appendix while keeping only main results in the paper.
    - Proof for theorem 4 is missing

*Patrick Chen, Wei-Cheng Chang, Jyun-Yu Jiang, Hsiang-Fu Yu, Inderjit Dhillon, and Cho-Jui Hsieh. 2023. FINGER: Fast Inference for Graph-based Approximate Nearest Neighbor Search. In Proceedings of the ACM Web Conference 2023 (WWW '23)*

Munoz, Javier Vargas, et al. "Hierarchical clustering-based graphs for large scale approximate nearest neighbor search." *Pattern Recognition*
 96 (2019): 106970.

**Questions:**

- Performance for NPG_{kgraph} on SIFT1M dataset is reported in Fig 5c, 6c and 7a. But the three graph do not seem to match.
- In Fig 5c), does NHQ-NPG_{kgraph} use proposed routing method and do other baselines (NHQ-HNSW and NHQ-NSG) use naive greedy search?
- What hyper-parameters were tried for ANNS + AF filtering approach?
- The proposed search strategy has two stages S1 and S2. How does transition from S1 to S2 happen?
- Complexity in Sec 4.1 : If C(u_i) is selected through an additional index then it will not be a constant time operation. Also in the final results, is C(u_i) selected randomly or using some additional index (and what index)?
- Some suggestion of writing and presentation
    - Including gridlines in the plot can help reading the graph.

**Limitations:**

No discussion

---

> ### Author Rebuttal · Authors · 2023-08-09
>
> **W1: Missing Baselines/Ablations**
>
> We appreciate your suggestions and include the missing ablations to validate our edge selection and routing strategies.
>
> * Graph construction strategy
>
> We compare our edge selection strategy with four existing ones: NGT [9], HNSW [6], HCNNG [10], and NSG [11] on SIFT1M and GIST1M datasets using a recent evaluation framework [5]. All competitors use the same test-time inference setup and distance function. We measure the Speedup-Recall metric, where Speedup is relative to brute force. Fig. 2 (a) and (b) in the attached PDF show that our strategy outperforms the others. For example, at Recall@10=0.9 on SIFT1M, our strategy achieves 1.1x, 1.4x, and 25.9x speedup over HCNNG/NSG, HNSW, and NGT, respectively. We also compare the index construction time and size of our strategy with the others in Tab. 3 of the PDF. Our strategy is faster and smaller than the others, demonstrating its superior efficiency and cost.
>
> * Graph navigation strategy
>
> We compare our graph navigation strategy with three existing ones: TOGG [12], FINGER [13], and HCNNG [10] within the HNSW index framework with consistent parameters. Tab. 4 in the PDF shows the speedup of each optimized strategy over the original HNSW at Recall@10 = 0.9 on SIFT1M. All optimized strategies are faster than the original HNSW, indicating the benefit of optimizing graph navigation. Our strategy has the lowest storage cost among the optimized strategies, as it does not require extra structures unlike the others. For instance, FINGER's storage cost is 3x higher than the original HNSW. This confirms the efficacy of our strategy, which improves search performance and minimizes storage requirements.
>
> **W2: Presentation Issue**
>
> Thank you for your suggestions. We'll move the pseudocodes of the composite index and joint search from the appendix to the main text and reorganize Section 3 for better clarity. Additionally, we'll include the pseudocodes of our edge selection and routing in the final version.
>
> We appreciate you pointing out the issue of poor readability due to important details being placed in the appendix. To address this, we'll move some crucial details (such as the experiment setup, datasets, and baseline description) into the main paper. Furthermore, we'll improve the hybrid distance section to make it more concise, allowing us to add these important details. Additionally, we'll transfer extra results from our experiments to the appendix, while retaining only the main results in the paper.
>
> We have double checked our appendix, and the proof for theorem 4 can be found in Appendix L (bottom of page 18 in our appendix).
>
> **Q1**
>
> In Fig. 5(c) and 6(c), we present the performance of the hybrid query (HQ) on the SIFT1M dataset. The difference in the scales of the axes makes the performance curves of NHQ-$NPG_{kgraph}$ appear mismatched, but they are actually identical in both figures. In Fig. 7(a), we report the performance of ANNS on SIFT1M. Therefore, it is natural that the QPS-Recall curve for $NPG_{kgraph}$ in Fig. 7(a) does not match the QPS-Recall curves for NHQ-$NPG_{kgraph}$ in Fig. 5(c) and 6(c).
>
> **Q2**
>
> Yes, in Fig. 5(c), NHQ-$NPG_{kgraph}$ utilizes the proposed routing method, while the other baselines (NHQ-HNSW and NHQ-NSG) employ naive greedy search. We'll clarify this point in the revision.
>
> **Q3**
>
> In the ANNS+AF filtering approaches, there are two types of hyperparameters involved.
>
> The first type pertains to the parameters specific to the ANNS methods used. We determine the optimal parameter configuration for these methods by referring to their respective papers or repositories (if provided) or by performing a grid search (if not provided). For example, parameters such as the maximum number of neighbors ($M$) and the size of the candidate set when selecting neighbors ($ef_{construction}$) are considered for HNSW.
>
> The second type of hyperparameter relates to the intermediate result size ($C$), where the intermediate results are obtained from the ANNS methods. We apply the AF on these intermediate results to produce the final hybrid query results. Thus, $C$ impacts both search efficiency and accuracy. Since the optimal $C$ value may vary depending on the dataset and queries, it's challenging to predict it in advance. Therefore, we conduct a grid search to determine the appropriate $C$ before conducting the query tests. It's important to note that we generate different QPS-Recall pairs by adjusting the search candidate size of the ANNS methods. For example, we modify the parameter $ef_{search}$ in HNSW to yield different QPS-Recall trade-offs.
>
> We'll include the hyper-parameter discussion of all methods in our appendix.
>
> **Q4**
>
> The transition from stage S1 to S2 in the proposed routing strategy occurs when stage S1 reaches a local optimum, indicated by the candidate set $R$ not being updated. In stage S2, we proceed to update the candidate set $R$ by checking all neighbors of the visited vertex. The search process terminates when the candidate set $R$ no longer receives any further updates.
>
> We'll add a pseudocode of our routing strategy to clarify it in the revision.
>
> **Q5**
>
> It's correct that if $C(u_i)$ is selected through an additional index, it will not be a constant time operation. We highlight that the complexity analysis in Section 4.1 of our paper is on the basis of a given $C(u_i)$. The total complexity analysis should consider the time complexity of obtaining $C(u_i)$. In the final version, we'll make sure to clarify this point.
>
> As stated in Section 4.3 of our paper, our edge selection process relies on two basal proximity graphs: NSW and KGraph. We obtain $C(u_i)$ using these two graph indexes (please refer to Appendix M for more details). The time complexity of obtaining $C(u_i)$ is $O(|V| \cdot \log^2(|V|))$ and $O(|V|^{1.14})$ for NSW and KGraph [5], respectively.
>
> **Q6**
>
> Thank you for the suggestion. We'll incorporate gridlines in all the plots to enhance readability.

---

> > ### Comment · Reviewer_ffbB · 2023-08-13
> > **Acknowledging author response and updating score**
> >
> > I have read the author response and my concerns have been addressed.
> > Having skimmed through other reviews and corresponding author responses, it looks like the authors have addressed major weaknesses pointed out by other reviewers.
> >
> > I am leaning towards accepting the paper and have updated my rating from `5: borderline accept` to `7: accept`.
> >
> > I would encourage authors to add these additional results to the paper (in the appendix perhaps) and improve the presentation of the results. The main set of results which answer key research questions can be in the main paper while other exhaustive set of results can be moved to the appendix.

---

> > > ### Author Response · Authors · 2023-08-14
> > > **Thanks! Will Add Additional Results.**
> > >
> > > Thank you for your prompt reply! We are glad that our responses helped clarify things. We are also grateful for your updated rating and the valuable feedback provided. We will include the additional results and enhance the presentation of our findings as you suggested.

---

### Official Review · Reviewer_MN6o · 2023-07-06

**Soundness:** 3 good
**Presentation:** 3 good
**Contribution:** 3 good
**Rating:** 6
**Confidence:** 3

**Summary:**

The paper discusses how hybrid query finds objects that are both similar to a feature vector and match some structured attributes. However, existing methods handle ANNS and attribute filtering separately, leading to inefficiency and inaccuracy. The paper proposes a new efficient and robust framework called native hybrid query (NHQ) and two new navigable PGs (NPGs) with optimized edge selection and routing, which improve the overall ANNS performance.

**Strengths:**

1. Optimized edge selection and routing are proposed which is efficient for ANNS problems.

2. The authors perform a sufficient complexity analysis of the proposed method. This helps readers understand the superiority and limitations of the proposed method.

3. Many experiments have been conducted. The authors have conducted experiments on multiple datasets to show their superiority in terms of accuracy, efficiency, and memory usage.

**Weaknesses:**

1. Compared to PQ-based methods, the PGs need more storage in runtime. Thus, a discussion on storage cost theoretically and experimentally of NHQ with PQ-based methods is necessary. There is still an improvement space on trade-off strategy on storage and efficiency.

2. More related works (HQ-based methods and edge selection strategies) are needed.

3. Lack of results of NHQ without edge selection strategy in ablution study.

**Questions:**

1. Discuss the storage cost theoretically and experimentally of NHQ with PQ-based methods.

2. Include results of NHQ without edge selection strategy. This comparison can help readers understand the strengths of the proposed edge selection strategy.

3. Include more related works (HQ-based methods and edge selection strategies).

**Limitations:**

Please refer to Paper Weakness.

---

> ### Author Rebuttal · Authors · 2023-08-09
>
> **W1, Q1: Storage Cost Analysis**
>
> We appreciate your suggestions and agree that comparing the storage cost of NHQ and PQ-based methods is necessary.
>
> * Theoretically
>
> NHQ and PQ-based methods have the same attribute storage cost, so we only analyze their feature vector storage cost.
>
> PQ-based methods compress high-dimensional vectors into the Cartesian product of multiple sub-codebooks. Let a vector $x$ be split into $M$ sub-vectors $u_j$, $1\leq j \leq M$, of dimension $D^*=D/M$, where $D$ is a multiple of $M$. The sub-vectors are quantized separately using $M$ quantizers with $K$ centroids each. We need to store the $M \times K$ centroids,  i.e., $KMD^* = KD$ floating-point values ($4KD$ bytes). Each vector is compressed as the code length $L=M\log_2K$ ($L/8$ bytes). The storage cost of PQ on $N$ vectors is: $4KD + NL/8$ bytes.
>
> NHQ builds a graph index for $N$ vectors with at most $R$ neighbors each. We use the adjacency list (a vertex with $R$ neighbor ids) to store the index, costing $4R$ bytes. We also store each raw vector with $ZD$ bytes. The storage cost of NHQ on $N$ vectors is: $ZND + 4NR$ bytes.
>
> On SIFT1M dataset, where $D=128$, $Z=4$, PQ sets $K=256$, $M=32$, costing 32131072 bytes (31MB). NHQ sets $R=20$, costing 592000000 bytes (565MB).
>
> Note that PQ is often combined with IVF, known as IVFPQ, which adds a coarse step to find probing centroids near the query. This increases the storage cost of PQ-based methods depending on specific optimizations.
>
> * Experimentally
>
> We compare the storage cost of NHQ and PQ-based methods on eight datasets in Tab. 2 of the attached PDF.
>
> We observe that: (1) NHQ costs more than PQ-based methods, consistent with our theory; (2) different PQ-based methods have different costs due to extra structures for optimization; (3) the same method has different costs on different datasets due to different optimal parameter configurations.
>
> Notably, PQ-based methods have low accuracy due to compression loss. As shown in [7] [8] and our experiments (Fig. 6 (a)-(c) of our paper), PQ-based techniques sacrifice accuracy (<0.8 on most datasets) for cost saving. In contrast, NHQ achieves close to 1 recall with high efficiency. Therefore, we believe that there is room for improvement in the trade-off between storage and search performance. We plan to explore this in our future work.
>
> We'll add this analysis to our final version.
>
> **W2, W3, Q2, Q3: Missing Baselines/Ablations**
>
> We appreciate your suggestions and have added more related works and evaluations on HQ-based methods and edge selection strategies.
>
> - HQ-based methods
>
> We noticed that Filtered-DiskANN [3] was released after our submission. Before that, we had considered all related work on hybrid queries and compared all SOTA methods. Filtered-DiskANN proposes optimizations called FilteredVamana and StitchedVamana on DiskANN. FilteredVamana connects vertices with shared attributes. StitchedVamana builds separate graph indexes for each filter and overlays them. While these optimizations enhance performance, Filtered-DiskANN's limitation lies in its inability to handle queries with multiple attributes. In contrast, NHQ supports any attribute combination in a query.
>
> We compare NHQ and Filtered-DiskANN on SIFT1M dataset, considering vectors with 3 attribute types. To execute Filtered-DiskANN, we only test single-attribute queries for all competitors. To eliminate other factors, we implement NHQ on DiskANN (NHQ can be easily extended to the current graph index), named NHQ-DiskANN. We keep the same parameters in DiskANN. The results are in Fig. 1 of the PDF. NHQ-DiskANN outperforms Filtered-DiskANN consistently, in both memory and disk versions. Upon analysis, we observe that SIFT1M has up to 180 attribute combinations, which may challenge Filtered-DiskANN in building a high-quality graph index. Additionally, when # attributes is large, StitchedVamana will build many graph indexes, increasing the indexing cost. Moreover, FilteredVamana only considers one matched attribute between a vertex and its neighbors, which also limits its application in complex attribute combinations.
>
> - Edge selection strategies
>
> We add more related works on edge selection strategies, including NGT [9], HNSW [6], HCNNG [10], and NSG [11].
>
> NGT: It builds a KNNG by incrementally inserting vertices and obtaining their nearest neighbors through a greedy search. It then optimizes the distribution of neighbors for each vertex using an effective path adjustment strategy.
>
> HNSW: It generates a hierarchical graph, where the vertices on the upper-level graph are a subset of the lower-level graph. It not only selects the nearest neighbors for an inserted point but also considers the distribution of neighbors using a heuristic edge selection strategy.
>
> HCNNG: It divides the dataset into multiple hierarchical clusters, where all points in each cluster are connected through a minimum spanning tree.
>
> NSG: It deploys an edge selection strategy based on the monotonic relative neighborhood graph. It prunes edges by searching for candidate neighbors on a KGraph, which is a KNNG.
>
> We compare our edge selection with the above four strategies on SIFT1M and GIST1M. To ensure a fair comparison, we implement our edge selection on a recent evaluation framework [5], which has deployed all above four strategies. Notably, all competitors use the same routing strategy and distance function. Fig. 2 (a) and (b) in the PDF show our strategy outperforms all other competitors for the Speedup-Recall trade-off. Our strategy also demonstrates more efficient index construction and smaller index size in Tab. 3 of the PDF.
>
> Additionally, in Fig. 2 (c) and (d) of the PDF, we evaluate indexes with and without our edge selection strategy, keeping other design variables the same. The results show that our edge selection brings significant performance gains in both the context of HQ and ANNS.
>
> We'll summarize the above analysis and include the main results in the revision.

---

### Official Review · Reviewer_XZ58 · 2023-07-07

**Soundness:** 2 fair
**Presentation:** 2 fair
**Contribution:** 2 fair
**Rating:** 4
**Confidence:** 3

**Summary:**

Attribute filtering (AF) is an important part of many scenarios using nearest neighbor search. Here, each data points has a feature vector in a geometric space and also a set of attributes (e.g., data, author) and queries must be matched to nearest vectors satisfying some attribute constraints.

While many algorithms have been studied for the classic ANNS problem, ANNS + AF is hard and needs new algorithms. Recently, there has been a flurry of attempts at this. Some of the basic approaches include filtering results of classic ANNS (which tends be yield poor results) or building separate indices for each attribute (Which leads to duplication).

This paper proposed that a better way to address this might be to create a fusion distance that incorporates geometric distance between feature vectors and suitably normalized similary score between attribute vectors. They then argue that a proximity graph data structure can be built using this distance. And that this performs better than other baselines selected in the paper.

**Strengths:**

The authors identify and articulate an important problem.
Empirical comparisons are made with many baselines.

**Weaknesses:**

There is no description of the dataset design in the main section. In appendix O, the datasets are described as usual vector datasets with 3 attributes (e.g., date, location, size) It looks like each vector can only have one possible combination of the attributes. So one can model it is one attribute dimension (cross-product) In such a case, why it not be easier and faster to build separate indices for each possible choice of attributes? Isnt complex index design only needed when datasets have multiple labels with an attribute dimension? In any case, it is impossible to evaluate the algorithm without well motivated datasets.

Missing baseline: filter-diskann [www'23], methods therein and equivalent code (in-memory and on-disk) are not compared. Instead weaker baselines are compared.

There is far too much greek notation. The notation can be greatly simplified for better readability. Simple pseudocode would also make for much better reading.

**Questions:**

In preliminaries, can you clearly state the attribution filtering problem and the defintion of recall for attribute filtering problem? It is unclear how the attribute vectors or indexed data and queries are matched.


 Please describe the datasets clearly in the main section. Please describe the encoding l(.) function for your datasets.

why are proximity graphs (PG) and not navigation graphs (e.g., NSG, HNSW) not the starting points for your construction. Proximity graphs tend to be disconnected even for non-AF use cases. Would like to see some analysis of connectivity of PG on AF-datasets.

**Limitations:**

-

---

> ### Author Rebuttal · Authors · 2023-08-09
>
> **W1: Dataset Issue**
>
> We appreciate your comments. Our datasets are diverse in size, # attributes, dimensions (feature and attribute vectors), and domains (image, video, etc.). Each vector has 3-9 attribute types with various values, common in real-world scenarios [1] [2].
>
> A data point with 3 attribute types (date, location, size) may have a value for each type. We agree that 3 values form one combination. But we consider not only the full combination of 3 values, but also partial combinations of 2 or 1 values. This leads to many combination cases (up to 26244 in our dataset). Some queries seek the nearest vectors that match partial attribute constraints (e.g., date regardless of location or size). Modeling the combination as a single attribute dimension is infeasible for such queries. Building separate indexes for each combination is also time-consuming [3] and impractical for queries with partial constraints. Hence, we designed a composite index that can handle diverse queries based on fused distances (it computes the preferred attribute dimensions while ignoring the unconstrained ones).
>
> We’ll clarify the dataset design in the main text.
>
> **W2: Missing Baselines**
>
> Thanks for your suggestions and sorry for the missing baselines. We noticed that Filtered-DiskANN [3] was released after our submission. Before that, we had considered all related work on hybrid queries and compared all SOTA methods.
>
> Filtered-DiskANN proposes two optimizations based on DiskANN: FilteredVamana and StitchedVamana. FilteredVamana connects vertices with shared attributes. StitchedVamana builds separate graph indexes for each filter and overlays them. These optimizations improve performance significantly. However, Filtered-DiskANN only supports single-attribute queries. This limits its applicability in scenarios requiring multiple attributes. For example, in product search, users may input a query image with color and size filters. Filtered-DiskANN cannot handle such cases. In contrast, NHQ supports any attribute combination in a query.
>
> We now add a comparison between NHQ and Filtered-DiskANN on SIFT1M dataset. Each vector has 3 attribute types, with 6, 6, and 5 values respectively. To execute Filtered-DiskANN, we only test single-attribute queries for all competitors. To eliminate other factors, we implement NHQ on DiskANN (NHQ can be easily extended to the current graph index), named NHQ-DiskANN. We keep the same parameters in DiskANN. The results are in Fig. 1 of the attached PDF. NHQ-DiskANN outperforms Filtered-DiskANN, in both memory and disk versions. Upon analysis, we observe that SIFT1M has up to 180 attribute combinations, which may challenge Filtered-DiskANN in building a high-quality graph index. Additionally, when # attributes is large, StitchedVamana will build many graph indexes, increasing the indexing cost. Moreover, FilteredVamana only considers one matched attribute between a vertex and its neighbors, which also limits its application in complex attribute combinations.
>
> **W3: Presentation Issue**
>
> Thank you for your suggestions to enhance readability. We'll use simpler symbols instead of Greek notation and add more pseudocodes. We'll also move the pseudocodes of the composite index and joint search from the appendix to the main text. Moreover, we'll provide the pseudocodes of our edge selection and routing.
>
> **Q1**
>
> Thanks for raising this point. The AF problem involves an object set $C$ and a query object $q$ with attributes $a_1,\cdots,a_m$ of size $m$. The goal is to find objects $G$ in $C$ that share the same attributes as $q$. For any $e\in G$, it holds $\forall i=1,2, \cdots, m, e . a_{i}=q . a_{i}$, where $e . a_{i}$ is the value of attribute $a_i$ of $e$. Note that $q$ may have partial attribute combinations (see W1). In this case, $e$ can have any value for the unconstrained attribute types. For the ANNS+AF problem, we also require the feature vector of $e$ in $G$ to be closest to that of $q$, resulting in the ground-truth set $G^*$ of ANNS+AF. Hence, the recall for ANNS+AF is $\frac{|R\cap G^*|}{k}$, where $R$ is the result set from an ANNS+AF method, and $k$ is the result size. However, recall for AF alone is unnecessary, as AF requires all results to match $q$'s attribute constraint, leading to a recall of 1.
>
> To illustrate attribute matching, consider an example with 2 attribute types: date and city. Suppose we have an object $e$ with $e.date=2023$ and $e.city=New\ York$, encoded as an attribute vector [1,3]. When the attribute vector of a query $q$ is [1,3] or [1,null] (where 'null' means no constraint), the attribute distance between $e$ and $q$ is 0, indicating a perfect match. However, if $q$ is [2,1] or [null,1], the attribute distance is 2 since the attributes of $q$ and $e$ are mismatched.
>
> We'll clarify AF in preliminaries.
>
> **Q2**
>
> Thanks for your suggestions. We follow previous works [4] and use ordinal encoding (i.e., our l(.)) to obtain attribute vectors for all datasets. For example, “New York” is encoded as 1 and “Beijing” as 2. Note that l(.) can use other encoders such as one-hot encoding.
>
> We'll clarify the datasets and l(.) in the main text.
>
> **Q3**
>
> Most navigation graphs diversify the neighbors on a base graph. For example, NSG and HNSW implement their edge selection based on KGraph and NSW, respectively, and achieve SOTA performance for ANNS [5]. Similarly, we build our navigation graph based on KGraph and NSW, using our edge selection. The results show that our strategy outperforms NSG and HNSW (see Fig. 2 of the PDF).
>
> We evaluate the connectivity of PG on both AF-datasets and non-AF-datasets (see Tab. 1 in the PDF). The results show that PG has similar connectivity in both cases. We know that KNNG has poor connectivity due to the cluster characteristics of non-AF-datasets [5]. PG diversifies the neighbor distribution and connects clusters [6]. For AF-datasets, vectors in different clusters may have the same attributes, which helps connectivity in NHQ.

---

### Author Rebuttal · Authors · 2023-08-09

We would like to express our gratitude to all four referees for providing us with valuable suggestions regarding the presentation and experimental studies. These suggestions have been immensely helpful in enhancing the quality of our paper. In response to the major concerns raised, we have conducted additional key experiments and included the main results in the attached PDF file. Furthermore, we will carefully revise the methodology and experiment sections to enhance their readability in the final version. Please refer to the detailed responses provided below for each individual reviewer. Due to space limitations, we put all references in here.

**References**

[1] VLDB'20: Wei, et al. Analyticdb-v: A hybrid analytical engine towards query fusion for structured and unstructured data.

[2] SIGMOD'21: Wang, et al. Milvus: A purpose-built vector data management system.

[3] WWW'23: Gollapudi, et al. Filtered-DiskANN: Graph algorithms for approximate nearest neighbor search with filters.

[4] TPAMI'15: Wang, et al. Exploring local and overall ordinal information for robust feature description.

[5] VLDB'21: Wang, et al. A comprehensive survey and experimental comparison of graph-based approximate nearest neighbor search.

[6] TPAMI'18: Malkov, et al. Efficient and robust approximate nearest neighbor search using Hierarchical Navigable Small World graphs.

[7] NeurIPS'19: Jayaram Subramanya, et al. Diskann: Fast accurate billion-point nearest neighbor search on a single node.

[8] NeurIPS'20: Ren, et al. HM-ANN : Efficient billion-point nearest neighbor search on heterogeneous memory.

[9] SISAP'16: Iwasaki, et al. Pruned bi-directed k-nearest neighbor graph for proximity search.

[10] PR'19: Hierarchical clustering-based graphs for large scale approximate nearest neighbor search.

[11] VLDB'19: Fast approximate nearest neighbor search with the navigating spreading-out graph.

[12] KBS'21: Xu, et al. Two-stage routing with optimized guided search and greedy algorithm on proximity graph.

[13] WWW'23: Chen, et al.  FINGER: Fast inference for graph-based approximate nearest neighbor search.

---

### Comment · Area_Chair_5o9C · 2023-08-17
**More discussions on reviews and rebuttals**

Dear Reviewers and Authors,

Close discussions among reviewers and authors are important for a fair judgetment to a submission. Please go ahead for or continue the discussions！

AC

---

> ### Author Response · Authors · 2023-08-18
> **Looking Forward to Feedback**
>
> Dear AC and Reviewers,
>
> Thank you for your comments and for facilitating the discussions among reviewers and authors. We appreciate your efforts and dedication to ensure a fair and rigorous evaluation of our submission. We also would like to thank the reviewers' efforts and time in reviewing our work.
>
> We are happy to answer any questions or concerns that you or the reviewers may have regarding our paper. We are also open to receiving any constructive feedback or suggestions that can help improve the quality and clarity of our work.
>
> Please feel free to contact us at any time. We look forward to hearing from you soon.
>
> Best regards,
>
> The authors

---

### Decision · Program_Chairs · 2023-09-21

**Decision:**

Accept (poster)

**Comment:**

The paper introduces a novel approximate nearest neighbor search framework which baked in attribute constraints via a single composite index compared to many existing two stage solutions.

The reviewers are overall positive on the ideas and rebuttals. MN6o: satisfied with the experiments and sufficient complexity analysis. ffbB: After rebuttal, satisfied with the rebuttal on experiments and presentation, scored from 5 to 7. 7xuq: idea is excellent.

After the rebuttal, Reviewer XZ58 still has concerns on the presentation, but "Overall, I don't disagree with the ideas". After read the comments and discussions of others as well as the paper, I feel that the concerns are resolvable in the final version, and encourage the authors to refine as promised in the rebuttal.

Based on the paper, reviews, rebuttals and discussions, I would like to recommend to accept as Poster.